# SHARING KNOWLEDGE IN MULTI-TASK DEEP REINFORCEMENT LEARNING

**Carlo D'Eramo & Davide Tateo**
Department of Computer Science
TU Darmstadt, IAS
Hochschulstraße 10, 64289, Darmstadt, Germany
`{carlo.deramo,davide.tateo}@tu-darmstadt.de`

**Andrea Bonarini & Marcello Restelli**
Politecnico di Milano, DEIB
Piazza Leonardo da Vinci 32, 20133, Milano
`{andrea.bonarini,marcello.restelli}@polimi.it`

**Jan Peters**
TU Darmstadt, IAS
Hochschulstraße 10, 64289, Darmstadt, Germany
Max Planck Institute for Intelligent Systems
Max-Planck-Ring 4, 72076, Tübingen, Germany
`jan.peters@tu-darmstadt.de`

## ABSTRACT

We study the benefit of sharing representations among tasks to enable the effective use of deep neural networks in Multi-Task Reinforcement Learning. We leverage the assumption that learning from different tasks, sharing common properties, is helpful to generalize the knowledge of them resulting in a more effective feature extraction compared to learning a single task. Intuitively, the resulting set of features offers performance benefits when used by Reinforcement Learning algorithms. We prove this by providing theoretical guarantees that highlight the conditions for which is convenient to share representations among tasks, extending the well-known finite-time bounds of Approximate Value-Iteration to the multi-task setting. In addition, we complement our analysis by proposing multi-task extensions of three Reinforcement Learning algorithms that we empirically evaluate on widely used Reinforcement Learning benchmarks showing significant improvements over the single-task counterparts in terms of sample efficiency and performance.

## 1 INTRODUCTION

Multi-Task Learning (MTL) ambitiously aims to learn multiple tasks jointly instead of learning them separately, leveraging the assumption that the considered tasks have common properties which can be exploited by Machine Learning (ML) models to generalize the learning of each of them. For instance, the features extracted in the hidden layers of a neural network trained on multiple tasks have the advantage of being a general representation of structures common to each other. This translates into an effective way of learning multiple tasks at the same time, but it can also improve the learning of each individual task compared to learning them separately (Caruana, 1997). Furthermore, the learned representation can be used to perform Transfer Learning (TL), i.e. using it as a preliminary knowledge to learn a new similar task resulting in a more effective and faster learning than learning the new task from scratch (Baxter, 2000; Thrun & Pratt, 2012).

The same benefits of extraction and exploitation of common features among the tasks achieved in MTL, can be obtained in Multi-Task Reinforcement Learning (MTRL) when training a single agent on multiple Reinforcement Learning (RL) problems with common structures (Taylor & Stone, 2009; Lazaric, 2012). In particular, in MTRL an agent can be trained on multiple tasks in the same

domain, e.g. riding a bicycle or cycling while going towards a goal, or on different but similar domains, e.g. balancing a pendulum or balancing a double pendulum[1]. Considering recent advances in Deep Reinforcement Learning (DRL) and the resulting increase in the complexity of experimental benchmarks, the use of Deep Learning (DL) models, e.g. deep neural networks, has become a popular and effective way to extract common features among tasks in MTRL algorithms (Rusu et al., 2015; Liu et al., 2016; Higgins et al., 2017). However, despite the high representational capacity of DL models, the extraction of good features remains challenging. For instance, the performance of the learning process can degrade when unrelated tasks are used together (Caruana, 1997; Baxter, 2000); another detrimental issue may occur when the training of a single model is not balanced properly among multiple tasks (Hessel et al., 2018).

Recent developments in MTRL achieve significant results in feature extraction by means of algorithms specifically developed to address these issues. While some of these works rely on a single deep neural network to model the multi-task agent (Liu et al., 2016; Yang et al., 2017; Hessel et al., 2018; Wulfmeier et al., 2019), others use multiple deep neural networks, e.g. one for each task and another for the multi-task agent (Rusu et al., 2015; Parisotto et al., 2015; Higgins et al., 2017; Teh et al., 2017). Intuitively, achieving good results in MTRL with a single deep neural network is more desirable than using many of them, since the training time is likely much less and the whole architecture is easier to implement. In this paper we study the benefits of shared representations among tasks. We theoretically motivate the intuitive effectiveness of our method, deriving theoretical guarantees that exploit the theoretical framework provided by Maurer et al. (2016), in which the authors present upper bounds on the quality of learning in MTL when extracting features for multiple tasks in a single shared representation. The significancy of this result is that the cost of learning the shared representation decreases with a factor $\mathcal{O}(1/\sqrt{T})$, where $T$ is the number of tasks for many function approximator hypothesis classes. The main *contribution* of this work is twofold.

1. We derive upper confidence bounds for Approximate Value-Iteration (AVI) and Approximate Policy-Iteration (API)[2] (Farahmand, 2011) in the MTRL setting, and we extend the approximation error bounds in Maurer et al. (2016) to the case of multiple tasks with different dimensionalities. Then, we show how to combine these results resulting in, to the best of our knowledge, the first proposed extension of the finite-time bounds of AVI/API to MTRL. Despite being an extension of previous works, we derive these results to justify our approach showing how the error propagation in AVI/API can theoretically benefit from learning multiple tasks jointly.

2. We leverage these results proposing a neural network architecture, for which these bounds hold with minor assumptions, that allow us to learn multiple tasks with a single regressor extracting a common representation. We show an empirical evidence of the consequence of our bounds by means of a variant of Fitted $Q$-Iteration (FQI) (Ernst et al., 2005), based on our shared network and for which our bounds apply, that we call Multi Fitted $Q$-Iteration (MFQI). Then, we perform an empirical evaluation in challenging RL problems proposing multi-task variants of the Deep $Q$-Network (DQN) (Mnih et al., 2015) and Deep Deterministic Policy Gradient (DDPG) (Lillicrap et al., 2015) algorithms. These algorithms are practical implementations of the more general AVI/API framework, designed to solve complex problems. In this case, the bounds apply to these algorithms only with some assumptions, e.g. stationary sampling distribution. The outcome of the empirical analysis joins the theoretical results, showing significant performance improvements compared to the single-task version of the algorithms in various RL problems, including several MuJoCo (Todorov et al., 2012) domains.

## 2 PRELIMINARIES

Let $B(\mathcal{X})$ be the space of bounded measurable functions w.r.t. the $\sigma$-algebra $\sigma_{\mathcal{X}}$, and similarly $B(\mathcal{X}, L)$ be the same bounded by $L < \infty$.

A Markov Decision Process (MDP) is defined as a 5-tuple $\mathcal{M} = <\mathcal{S}, \mathcal{A}, \mathcal{P}, \mathcal{R}, \gamma>$, where $\mathcal{S}$ is the state space, $\mathcal{A}$ is the action space, $\mathcal{P} : \mathcal{S} \times \mathcal{A} \to \mathcal{S}$ is the transition distribution where $\mathcal{P}(s'|s, a)$

---

[1]For simplicity, in this paper we refer to the concepts of *task* and *domain* interchangeably.
[2]All proofs and the theorem for API are in Appendix A.2.

is the probability of reaching state $s'$ when performing action $a$ in state $s$, $\mathcal{R} : \mathcal{S} \times \mathcal{A} \times \mathcal{S} \to \mathbb{R}$ is the reward function, and $\gamma \in (0, 1]$ is the discount factor. A *deterministic policy* $\pi$ maps, for each state, the action to perform: $\pi : \mathcal{S} \to \mathcal{A}$. Given a policy $\pi$, the value of an action $a$ in a state $s$ represents the expected discounted cumulative reward obtained by performing $a$ in $s$ and following $\pi$ thereafter: $Q^\pi(s, a) \triangleq \mathbb{E}[\sum_{k=0}^\infty \gamma^k r_{i+k+1}|s_i = s, a_i = a, \pi]$, where $r_{i+1}$ is the reward obtained after the $i$-th transition. The expected discounted cumulative reward is maximized by following the *optimal* policy $\pi^*$ which is the one that determines the optimal action values, i.e., the ones that satisfy the Bellman optimality equation (Bellman, 1954): $Q^*(s, a) \triangleq \int_\mathcal{S} \mathcal{P}(s'|s, a) [\mathcal{R}(s, a, s') + \gamma \max_{a'} Q^*(s', a')] \, ds'$. The solution of the Bellman optimality equation is the fixed point of the optimal Bellman operator $\mathcal{T}^* : B(\mathcal{S} \times \mathcal{A}) \to B(\mathcal{S} \times \mathcal{A})$ defined as $(\mathcal{T}^* Q)(s, a) \triangleq \int_\mathcal{S} \mathcal{P}(s'|s, a)[\mathcal{R}(s, a, s') + \gamma \max_{a'} Q(s', a')] ds'$. In the MTRL setting, there are multiple MDPs $\mathcal{M}^{(t)} = < \mathcal{S}^{(t)}, \mathcal{A}^{(t)}, \mathcal{P}^{(t)}, \mathcal{R}^{(t)}, \gamma^{(t)} >$ where $t \in \{1, \dots, T\}$ and $T$ is the number of MDPs. For each MDP $\mathcal{M}^{(t)}$, a deterministic policy $\pi_t : \mathcal{S}^{(t)} \to \mathcal{A}^{(t)}$ induces an action-value function $Q_t^{\pi_t}(s^{(t)}, a^{(t)}) = \mathbb{E}[\sum_{k=0}^\infty \gamma^k r_{i+k+1}^{(t)}|s_i = s^{(t)}, a_i = a^{(t)}, \pi_t]$. In this setting, the goal is to maximize the sum of the expected cumulative discounted reward of each task.

In our theoretical analysis of the MTRL problem, the complexity of representation plays a central role. As done in Maurer et al. (2016), we consider the Gaussian complexity, a variant of the well-known Rademacher complexity, to measure the complexity of the representation. Given a set $\bar{\mathbf{X}} \in \mathcal{X}^{Tn}$ of $n$ input samples for each task $t \in \{1, \dots, T\}$, and a class $\mathcal{H}$ composed of $k \in \{1, \dots, K\}$ functions, the Gaussian complexity of a random set $\mathcal{H}(\bar{\mathbf{X}}) = \{(h_k(X_{ti})) : h \in \mathcal{H}\} \subseteq \mathbb{R}^{KTn}$ is defined as follows:

$$G(\mathcal{H}(\bar{\mathbf{X}})) = \mathbb{E}\left[\sup_{h \in \mathcal{H}} \sum_{tki} \gamma_{tki} h_k(X_{ti}) \middle| X_{ti}\right], \tag{1}$$

where $\gamma_{tki}$ are independent standard normal variables. We also need to define the following quantity, taken from Maurer (2016): let $\boldsymbol{\gamma}$ be a vector of $m$ random standard normal variables, and $f \in \mathcal{F} : Y \to \mathbb{R}^m$, with $Y \subseteq \mathbb{R}^n$, we define

$$O(\mathcal{F}) = \sup_{y,y' \in Y, y \neq y'} \mathbb{E}\left[\sup_{f \in \mathcal{F}} \frac{\langle \boldsymbol{\gamma}, f(y) - f(y')\rangle}{\|y - y'\|}\right]. \tag{2}$$

Equation 2 can be viewed as a Gaussian average of Lipschitz quotients, and appears in the bounds provided in this work. Finally, we define $L(\mathcal{F})$ as the upper bound of the Lipschitz constant of all the functions $f$ in the function class $\mathcal{F}$.

# 3 THEORETICAL ANALYSIS

The following theoretical study starts from the derivation of theoretical guarantees for MTRL in the AVI framework, extending the results of Farahmand (2011) in the MTRL scenario. Then, to bound the approximation error term in the AVI bound, we extend the result described in Maurer (2006) to MTRL. As we discuss, the resulting bounds described in this section clearly show the benefit of sharing representation in MTRL. To the best of our knowledge, this is the first general result for MTRL; previous works have focused on finite MDPs (Brunskill & Li, 2013) or linear models (Lazaric & Restelli, 2011).

## 3.1 MULTI-TASK REPRESENTATION LEARNING

The multi-task representation learning problem consists in learning simultaneously a set of $T$ tasks $\mu_t$, modeled as probability measures over the space of the possible input-output pairs $(x, y)$, with $x \in \mathcal{X}$ and $y \in \mathbb{R}$, being $\mathcal{X}$ the input space. Let $w \in \mathcal{W} : \mathcal{X} \to \mathbb{R}^J$, $h \in \mathcal{H} : \mathbb{R}^J \to \mathbb{R}^K$ and $f \in \mathcal{F} : \mathbb{R}^K \to \mathbb{R}$ be functions chosen from their respective hypothesis classes. The functions in the hypothesis classes must be Lipschitz continuous functions. Let $\bar{\mathbf{Z}} = (\mathbf{Z}_1, \dots, \mathbf{Z}_T)$ be the multi-sample over the set of tasks $\boldsymbol{\mu} = (\mu_1, \dots, \mu_T)$, where $\mathbf{Z}_t = (Z_{t1}, \dots, Z_{tn}) \sim \mu_t^n$ and $Z_{ti} = (X_{ti}, Y_{ti}) \sim \mu_t$. We can formalize our regression problem as the following minimization

problem:

$$\min\left\{\frac{1}{nT}\sum_{t=1}^{T}\sum_{i=1}^{N}\ell(f_t(h(w_t(X_{ti}))),Y_{ti}) : \mathbf{f}\in\mathcal{F}^T, h\in\mathcal{H}, \mathbf{w}\in\mathcal{W}^T\right\}, \qquad (3)$$

where we use $\mathbf{f}=(f_1,\ldots,f_T)$, $\mathbf{w}=(w_1,\ldots,w_T)$, and define the minimizers of Equation (3) as $\hat{\mathbf{w}}$, $\hat{h}$, and $\hat{\mathbf{f}}$. We assume that the loss function $\ell:\mathbb{R}\times\mathbb{R}\rightarrow[0,1]$ is 1-Lipschitz in the first argument for every value of the second argument. While this assumption may seem restrictive, the result obtained can be easily scaled to the general case. To use the principal result of this section, for a generic loss function $\ell'$, it is possible to use $\ell(\cdot)=\ell'(\cdot)/\epsilon_{\max}$, where $\epsilon_{\max}$ is the maximum value of $\ell'$. The expected loss over the tasks, given $\mathbf{w}$, $h$ and $\mathbf{f}$ is the task-averaged risk:

$$\varepsilon_{\text{avg}}(\mathbf{w},h,\mathbf{f}) = \frac{1}{T}\sum_{t=1}^{T}\mathbb{E}\left[\ell(f_t(h(w_t(X))),Y)\right] \qquad (4)$$

The minimum task-averaged risk, given the set of tasks $\boldsymbol{\mu}$ and the hypothesis classes $\mathcal{W}$, $\mathcal{H}$ and $\mathcal{F}$ is $\varepsilon_{\text{avg}}^*$, and the corresponding minimizers are $\mathbf{w}^*$, $h^*$ and $\mathbf{f}^*$.

### 3.2 Multi-task Approximate Value Iteration bound

We start by considering the bound for the AVI framework which applies for the single-task scenario.

**Theorem 1.** *(Theorem 3.4 of Farahmand (2011)) Let $K$ be a positive integer, and $Q_{max}\leq\frac{R_{max}}{1-\gamma}$. Then for any sequence $(Q_k)_{k=0}^{K}\subset B(\mathcal{S}\times\mathcal{A},Q_{max})$ and the corresponding sequence $(\varepsilon_k)_{k=0}^{K-1}$, where $\varepsilon_k=\|Q_{k+1}-\mathcal{T}^*Q_k\|_{\nu}^2$, we have:*

$$\|Q^*-Q^{\pi_K}\|_{1,\rho}\leq\frac{2\gamma}{(1-\gamma)^2}\left[\inf_{r\in[0,1]}C_{VI,\rho,\nu}^{\frac{1}{2}}(K;r)\mathcal{E}^{\frac{1}{2}}(\varepsilon_0,\ldots,\varepsilon_{K-1};r)+\frac{2}{1-\gamma}\gamma^K R_{max}\right], \qquad (5)$$

*where*

$$C_{VI,\rho,\nu}(K;r) = \left(\frac{1-\gamma}{2}\right)^2\sup_{\pi_1',\ldots,\pi_K'}\sum_{k=0}^{K-1}a_k^{2(1-r)}\left[\sum_{m\geq0}\gamma^m\Big(c_{VI_1,\rho,\nu}(m,K-k;\pi_K')\right.$$

$$\left.+c_{VI_2,\rho,\nu}(m+1;\pi_{k+1}',\ldots,\pi_K')\Big)\right]^2, \qquad (6)$$

*with $\mathcal{E}(\varepsilon_0,\ldots,\varepsilon_{K-1};r)=\sum_{k=0}^{K-1}\alpha_k^{2r}\varepsilon_k$, the two coefficients $c_{VI_1,\rho,\nu}$, $c_{VI_2,\rho,\nu}$, the distributions $\rho$ and $\nu$, and the series $\alpha_k$ are defined as in Farahmand (2011).*

In the multi-task scenario, let the average approximation error across tasks be:

$$\varepsilon_{\text{avg},k}(\hat{\mathbf{w}}_k,\hat{h}_k,\hat{\mathbf{f}}_k) = \frac{1}{T}\sum_{t=1}^{T}\|Q_{t,k+1}-\mathcal{T}_t^*Q_{t,k}\|_{\nu}^2, \qquad (7)$$

where $Q_{t,k+1}=\hat{f}_{t,k}\circ\hat{h}_k\circ\hat{w}_{t,k}$, and $\mathcal{T}_t^*$ is the optimal Bellman operator of task $t$.

In the following, we extend the AVI bound of Theorem 1 to the multi-task scenario, by computing the average loss across tasks and pushing inside the average using Jensen's inequality.

**Theorem 2.** *Let $K$ be a positive integer, and $Q_{max}\leq\frac{R_{max}}{1-\gamma}$. Then for any sequence $(Q_k)_{k=0}^{K}\subset B(\mathcal{S}\times\mathcal{A},Q_{max})$ and the corresponding sequence $(\varepsilon_{avg,k})_{k=0}^{K-1}$, where $\varepsilon_{avg,k}=\frac{1}{T}\sum_{t=1}^{T}\|Q_{t,k+1}-\mathcal{T}_t^*Q_{t,k}\|_{\nu}^2$, we have:*

$$\frac{1}{T}\sum_{t=1}^{T}\|Q_t^*-Q_t^{\pi_K}\|_{1,\rho}\leq\frac{2\gamma}{(1-\gamma)^2}\left[\inf_{r\in[0,1]}C_{VI}^{\frac{1}{2}}(K;r)\mathcal{E}_{avg}^{\frac{1}{2}}(\varepsilon_{avg,0},\ldots,\varepsilon_{avg,K-1};r)+\frac{2\gamma^K R_{max,avg}}{1-\gamma}\right] \qquad (8)$$

*with $\mathcal{E}_{avg}=\sum_{k=0}^{K-1}\alpha_k^{2r}\varepsilon_{avg,k}$, $\gamma=\max_{t\in\{1,\ldots,T\}}\gamma_t$, $C_{VI}^{\frac{1}{2}}(K;r)=\max_{t\in\{1,\ldots,T\}}C_{VI,\rho,\nu}^{\frac{1}{2}}(K;t,r)$, $R_{max,avg}=\frac{1}{T}\sum_{t=1}^{T}R_{max,t}$ and $\alpha_k=\begin{cases}\frac{(1-\gamma)\gamma^{K-k-1}}{1-\gamma^{K+1}} & 0\leq k<K,\\\frac{(1-\gamma)\gamma^K}{1-\gamma^{K+1}} & k=K\end{cases}$.*

**Remarks** Theorem 2 retains most of the properties of Theorem 3.4 of Farahmand (2011), except that the regression error in the bound is now task-averaged. Interestingly, the second term of the sum in Equation (8) depends on the average maximum reward for each task. In order to obtain this result we use an overly pessimistic bound on $\gamma$ and the concentrability coefficients, however this approximation is not too loose if the MDPs are sufficiently similar.

### 3.3 Multi-task approximation error bound

We bound the task-averaged approximation error $\varepsilon_{\text{avg}}$ at each AVI iteration $k$ involved in (8) following a derivation similar to the one proposed by Maurer et al. (2016), obtaining:

**Theorem 3.** *Let $\boldsymbol{\mu}$, $\mathcal{W}$, $\mathcal{H}$ and $\mathcal{F}$ be defined as above and assume $0 \in \mathcal{H}$ and $f(0) = 0, \forall f \in \mathcal{F}$. Then for $\delta > 0$ with probability at least $1 - \delta$ in the draw of $\bar{\mathbf{Z}} \sim \prod_{t=1}^{T} \mu_t^n$ we have that*

$$\varepsilon_{avg}(\hat{\mathbf{w}}, \hat{h}, \hat{\mathbf{f}}) \leq L(\mathcal{F}) \left( c_1 \frac{L(\mathcal{H}) \sup_{l \in \{1,...,T\}} G(\mathcal{W}(\mathbf{X}_l))}{n} + c_2 \frac{\sup_{\mathbf{w}} \|\mathbf{w}(\bar{\mathbf{X}})\| O(\mathcal{H})}{nT} \right.$$

$$\left. + c_3 \frac{\min_{p \in P} G(\mathcal{H}(p))}{nT} \right) + c_4 \frac{\sup_{h,\mathbf{w}} \|h(\mathbf{w}(\bar{\mathbf{X}}))\| O(\mathcal{F})}{n\sqrt{T}} + \sqrt{\frac{8 \ln(\frac{3}{\delta})}{nT}} + \varepsilon_{avg}^*. \quad (9)$$

**Remarks** The assumptions $0 \in \mathcal{H}$ and $f(0) = 0$ for all $f \in \mathcal{F}$ are not essential for the proof and are only needed to simplify the result. For reasonable function classes, the Gaussian complexity $G(\mathcal{W}(\mathbf{X}_l))$ is $\mathcal{O}(\sqrt{n})$. If $\sup_{\mathbf{w}} \|\mathbf{w}(\bar{\mathbf{X}})\|$ and $\sup_{h,\mathbf{w}} \|h(\mathbf{w}(\bar{\mathbf{X}}))\|$ can be uniformly bounded, then they are $\mathcal{O}(\sqrt{nT})$. For some function classes, the Gaussian average of Lipschitz quotients $O(\cdot)$ can be bounded independently from the number of samples. Given these assumptions, the first and the fourth term of the right hand side of Equation (9), which represent respectively the cost of learning the meta-state space $\mathbf{w}$ and the task-specific $\mathbf{f}$ mappings, are both $\mathcal{O}(1/\sqrt{n})$. The second term represents the cost of learning the multi-task representation $h$ and is $\mathcal{O}(1/\sqrt{nT})$, thus vanishing in the multi-task limit $T \to \infty$. The third term can be removed if $\forall h \in \mathcal{H}, \exists p_0 \in P : h(p) = 0$; even when this assumption does not hold, this term can be ignored for many classes of interest, e.g. neural networks, as it can be arbitrarily small.

The last term to be bounded in (9) is the minimum average approximation error $\varepsilon_{\text{avg}}^*$ at each AVI iteration $k$. Recalling that the task-averaged approximation error is defined as in (7), applying Theorem 5.3 by Farahmand (2011) we obtain:

**Lemma 4.** *Let $Q_{t,k}^*, \forall t \in \{1, \ldots, T\}$ be the minimizers of $\varepsilon_{avg,k}^*$, $\check{t}_k = \arg\max_{t \in \{1,...,T\}} \|Q_{t,k+1}^* - \mathcal{T}_t^* Q_{t,k}\|_\nu^2$, and $b_{k,i} = \|Q_{\check{t}_k,i+1}^* - \mathcal{T}_{\check{t}}^* Q_{\check{t}_k,i}\|_\nu$, then:*

$$\varepsilon_{avg,k}^* \leq \left( \|Q_{\check{t}_k,k+1}^* - (\mathcal{T}_{\check{t}}^*)^{k+1} Q_{\check{t}_k,0}\|_\nu + \sum_{i=0}^{k-1} (\gamma_{\check{t}_k} C_{AE}(\nu; \check{t}_k, P))^{i+1} b_{k,k-1-i} \right)^2, \quad (10)$$

*with $C_{AE}$ defined as in Farahmand (2011).*

**Final remarks** The bound for MTRL is derived by composing the results in Theorems 2 and 3, and Lemma 4. The results above highlight the advantage of learning a shared representation. The bound in Theorem 2 shows that a small approximation error is critical to improve the convergence towards the optimal action-value function, and the bound in Theorem 3 shows that the cost of learning the shared representation at each AVI iteration is mitigated by using multiple tasks. This is particularly beneficial when the feature representation is complex, e.g. deep neural networks.

### 3.4 Discussion

As stated in the remarks of Equation (9), the benefit of MTRL is evinced by the second component of the bound, i.e. the cost of learning $h$, which vanishes with the increase of the number of tasks. Obviously, adding more tasks require the shared representation to be large enough to include all of them, undesirably causing the term $\sup_{h,\mathbf{w}} \|h(\mathbf{w}(\bar{\mathbf{X}}))\|$ in the fourth component of the bound to increase. This introduces a tradeoff between the number of features and number of tasks; however, for

| (a) Shared network | (b) FQI vs MFQI | (c) #Task analysis |

Figure 1: (a) The architecture of the neural network we propose to learn $T$ tasks simultaneously. The $w_t$ block maps each input $x_t$ from task $\mu_t$ to a *shared* set of layers $h$ which extracts a common representation of the tasks. Eventually, the shared representation is specialized in block $f_t$ and the output $y_t$ of the network is computed. Note that each block can be composed of arbitrarily many layers. (b) Results of FQI and MFQI averaged over $4$ tasks in *Car-On-Hill*, showing $\|Q^* - Q^{\pi_K}\|$ on the left, and the discounted cumulative reward on the right. (c) Results of MFQI showing $\|Q^* - Q^{\pi_K}\|$ for increasing number of tasks. Both results in (b) and (c) are averaged over $100$ experiments, and show the $95\%$ confidence intervals.

a reasonable number of tasks the number of features used in the single-task case is enough to handle them, as we show in some experiments in Section 5. Notably, since the AVI/API framework provided by Farahmand (2011) provides an easy way to include the approximation error of a generic function approximator, it is easy to show the benefit in MTRL of the bound in Equation (9). Despite being just multi-task extensions of previous works, our results are the first one to theoretically show the benefit of sharing representation in MTRL. Moreover, they serve as a significant theoretical motivation, besides to the intuitive ones, of the practical algorithms that we describe in the following sections.

## 4    SHARING REPRESENTATIONS

We want to empirically evaluate the benefit of our theoretical study in the problem of jointly learning $T$ different tasks $\mu_t$, introducing a neural network architecture for which our bounds hold. Following our theoretical framework, the network we propose extracts representations $w_t$ from inputs $x_t$ for each task $\mu_t$, mapping them to common features in a set of shared layers $h$, specializing the learning of each task in respective separated layers $f_t$, and finally computing the output $y_t = (f_t \circ h \circ w_t)(x_t) = f_t(h(w_t(x_t)))$ (Figure 1(a)). The idea behind this architecture is not new in the literature. For instance, similar ideas have already been used in DQN variants to improve exploration on the same task via bootstrapping (Osband et al., 2016) and to perform MTRL (Liu et al., 2016).

The intuitive and desirable property of this architecture is the exploitation of the regularization effect introduced by the shared representation of the jointly learned tasks. Indeed, unlike learning a single task that may end up in overfitting, forcing the model to compute a shared representation of the tasks helps the regression process to extract more general features, with a consequent reduction in the variance of the learned function. This intuitive justification for our approach, joins the theoretical benefit proven in Section 3. Note that our architecture can be used in any MTRL problem involving a regression process; indeed, it can be easily used in value-based methods as a $Q$-function regressor, or in policy search as a policy regressor. In both cases, the targets are learned for each task $\mu_t$ in its respective output block $f_t$. Remarkably, as we show in the experimental Section 5, it is straightforward to extend RL algorithms to their multi-task variants only through the use of the proposed network architecture, without major changes to the algorithms themselves.

## 5    EXPERIMENTAL RESULTS

To empirically evince the effect described by our bounds, we propose an extension of FQI (Ernst et al., 2005; Riedmiller, 2005), that we call MFQI, for which our AVI bounds apply. Then, to empirically evaluate our approach in challenging RL problems, we introduce multi-task variants of two well-known DRL algorithms: DQN (Mnih et al., 2015) and DDPG (Lillicrap et al., 2015), which we call Multi Deep $Q$-Network (MDQN) and Multi Deep Deterministic Policy Gradient (MDDPG) respectively. Note that for these methodologies, our AVI and API bounds hold only with

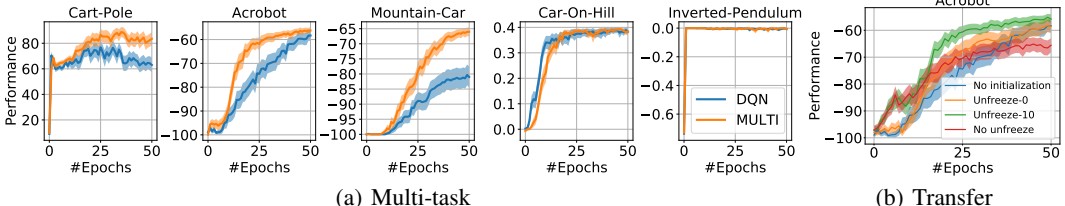

(a) Multi-task                                                    (b) Transfer

Figure 2: Discounted cumulative reward averaged over 100 experiments of DQN and MDQN for each task and for transfer learning in the *Acrobot* problem. An epoch consists of $1,000$ steps, after which the greedy policy is evaluated for $2,000$ steps. The $95\%$ confidence intervals are shown.

the simplifying assumption that the samples are i.i.d.; nevertheless they are useful to show the benefit of our method also in complex scenarios, e.g. MuJoCo (Todorov et al., 2012). We remark that in these experiments we are only interested in showing the benefit of learning multiple tasks with a shared representation w.r.t. learning a single task; therefore, we only compare our methods with the single task counterparts, ignoring other works on MTRL in literature. Experiments have been developed using the MushroomRL library (D'Eramo et al., 2020), and run on an NVIDIA® DGX Station™ and Intel® AI DevCloud. Refer to Appendix B for all the details and our motivations about the experimental settings.

## 5.1 MULTI FITTED $Q$-ITERATION

As a first empirical evaluation, we consider FQI, as an example of an AVI algorithm, to show the effect described by our theoretical AVI bounds in experiments. We consider the *Car-On-Hill* problem as described in Ernst et al. (2005), and select four different tasks from it changing the mass of the car and the value of the actions (details in Appendix B). Then, we run separate instances of FQI with a single task network for each task respectively, and one of MFQI considering all the tasks simultaneously. Figure 1(b) shows the $L_1$-norm of the difference between $Q^*$ and $\hat{Q}^{\pi_K}$ averaged over all the tasks. It is clear how MFQI is able to get much closer to the optimal $Q$-function, thus giving an empirical evidence of the AVI bounds in Theorem 2. For completeness, we also show the advantage of MFQI w.r.t. FQI in performance. Then, in Figure 1(c) we provide an empirical evidence of the benefit of increasing the number of tasks in MFQI in terms of both quality and stability.

## 5.2 MULTI DEEP $Q$-NETWORK

As in Liu et al. (2016), our MDQN uses separate replay memories for each task and the batch used in each training step is built picking the same number of samples from each replay memory. Furthermore, a step of the algorithm consists of exactly one step in each task. These are the only minor changes to the vanilla DQN algorithm we introduce, while all other aspects, such as the use of the target network, are not modified. Thus, the time complexity of MDQN is considerably lower than vanilla DQN thanks to the learning of $T$ tasks with a single model, but at the cost of a higher memory complexity for the collection of samples for each task. We consider five problems with similar state spaces, sparse rewards and discrete actions: *Cart-Pole*, *Acrobot*, *Mountain-Car*, *Car-On-Hill*, and *Inverted-Pendulum*. The implementation of the first three problems is the one provided by the OpenAI Gym library Brockman et al. (2016), while Car-On-Hill is described in Ernst et al. (2005) and Inverted-Pendulum in Lagoudakis & Parr (2003).

Figure 2(a) shows the performance of MDQN w.r.t. to vanilla DQN that uses a single-task network structured as the multi-task one in the case with $T = 1$. The first three plots from the left show good performance of MDQN, which is both higher and more stable than DQN. In Car-On-Hill, MDQN is slightly slower than DQN to reach the best performance, but eventually manages to be more stable. Finally, the Inverted-Pendulum experiment is clearly too easy to solve for both approaches, but it is still useful for the shared feature extraction in MDQN. The described results provide important hints about the better quality of the features extracted by MDQN w.r.t. DQN. To further demonstrate this, we evaluate the performance of DQN on Acrobot, arguably the hardest of the five problems, using a single-task network with the shared parameters in $h$ initialized with the weights of a multi-task

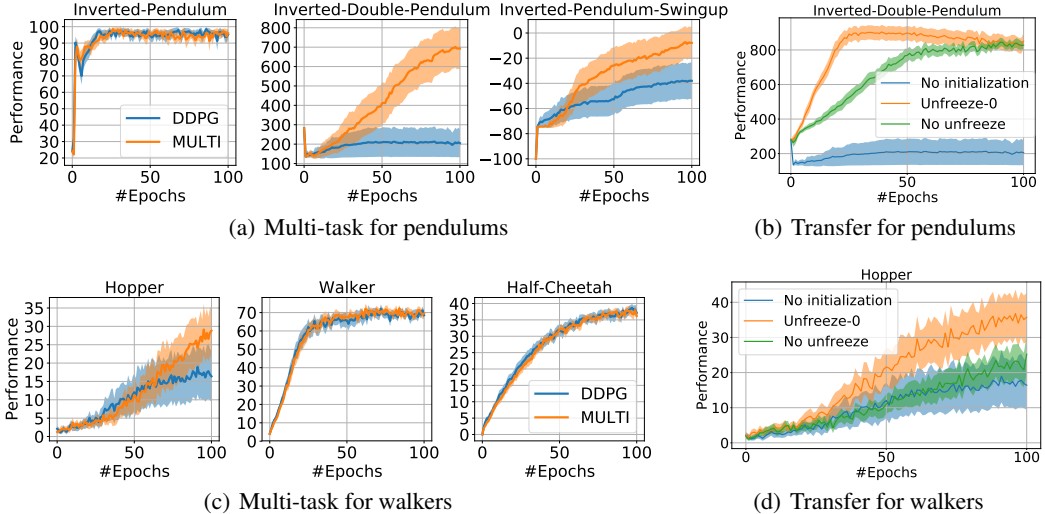

Figure 3: Discounted cumulative reward averaged over $40$ experiments of DDPG and MDDPG for each task and for transfer learning in the *Inverted-Double-Pendulum* and *Hopper* problems. An epoch consists of $10,000$ steps, after which the greedy policy is evaluated for $5,000$ steps. The $95\%$ confidence intervals are shown.

network trained with MDQN on the other four problems. Arbitrarily, the pre-trained weights can be adjusted during the learning of the new task or can be kept fixed and only the remaining randomly initialized parameters in **w** and **f** are trained. From Figure 2(b), the advantages of initializing the weights are clear. In particular, we compare the performance of DQN without initialization w.r.t. DQN with initialization in three settings: in *Unfreeze-0* the initialized weights are adjusted, in *No-Unfreeze* they are kept fixed, and in *Unfreeze-10* they are kept fixed until epoch $10$ after which they start to be optimized. Interestingly, keeping the shared weights fixed shows a significant performance improvement in the earliest epochs, but ceases to improve soon. On the other hand, the adjustment of weights from the earliest epochs shows improvements only compared to the uninitialized network in the intermediate stages of learning. The best results are achieved by starting to adjust the shared weights after epoch $10$, which is approximately the point at which the improvement given by the fixed initialization starts to lessen.

## 5.3 MULTI DEEP DETERMINISTIC POLICY GRADIENT

In order to show how the flexibility of our approach easily allows to perform MTRL in policy search algorithms, we propose MDDPG as a multi-task variant of DDPG. As an actor-critic method, DDPG requires an *actor* network and a *critic* network. Intuitively, to obtain MDDPG both the actor and critic networks should be built following our proposed structure. We perform separate experiments on two sets of MuJoCo Todorov et al. (2012) problems with similar continuous state and action spaces: the first set includes *Inverted-Pendulum*, *Inverted-Double-Pendulum*, and *Inverted-Pendulum-Swingup* as implemented in the *pybullet* library, whereas the second set includes *Hopper-Stand*, *Walker-Walk*, and *Half-Cheetah-Run* as implemented in the DeepMind Control SuiteTassa et al. (2018). Figure 3(a) shows a relevant improvement of MDDPG w.r.t. DDPG in the pendulum tasks. Indeed, while in Inverted-Pendulum, which is the easiest problem among the three, the performance of MDDPG is only slightly better than DDPG, the difference in the other two problems is significant. The advantage of MDDPG is confirmed in Figure 3(c) where it performs better than DDPG in Hopper and equally good in the other two tasks. Again, we perform a TL evaluation of DDPG in the problems where it suffers the most, by initializing the shared weights of a single-task network with the ones of a multi-task network trained with MDDPG on the other problems. Figures 3(b) and 3(d) show evident advantages of pre-training the shared weights and a significant difference between keeping them fixed or not.

## 6 RELATED WORKS

Our work is inspired from both theoretical and empirical studies in MTL and MTRL literature. In particular, the theoretical analysis we provide follows previous results about the theoretical properties of multi-task algorithms. For instance, Cavallanti et al. (2010) and Maurer (2006) prove the theoretical advantages of MTL based on linear approximation. More in detail, Maurer (2006) derives bounds on MTL when a linear approximator is used to extract a shared representation among tasks. Then, Maurer et al. (2016), which we considered in this work, describes similar results that extend to the use of non-linear approximators. Similar studies have been conducted in the context of MTRL. Among the others, Lazaric & Restelli (2011) and Brunskill & Li (2013) give theoretical proofs of the advantage of learning from multiple MDPs and introduces new algorithms to empirically support their claims, as done in this work.

Generally, contributions in MTRL assume that properties of different tasks, e.g. dynamics and reward function, are generated from a common generative model. About this, interesting analyses consider Bayesian approaches; for instance Wilson et al. (2007) assumes that the tasks are generated from a hierarchical Bayesian model, and likewise Lazaric & Ghavamzadeh (2010) considers the case when the value functions are generated from a common prior distribution. Similar considerations, which however does not use a Bayesian approach, are implicitly made in Taylor et al. (2007), Lazaric et al. (2008), and also in this work.

In recent years, the advantages of MTRL have been empirically evinced also in DRL, especially exploiting the powerful representational capacity of deep neural networks. For instance, Parisotto et al. (2015) and Rusu et al. (2015) propose to derive a multi-task policy from the policies learned by DQN experts trained separately on different tasks. Rusu et al. (2015) compares to a therein introduced variant of DQN, which is very similar to our MDQN and the one in Liu et al. (2016), showing how their method overcomes it in the Atari benchmark Bellemare et al. (2013). Further developments, extend the analysis to policy search (Yang et al., 2017; Teh et al., 2017), and to multi-goal RL (Schaul et al., 2015; Andrychowicz et al., 2017). Finally, Hessel et al. (2018) addresses the problem of balancing the learning of multiple tasks with a single deep neural network proposing a method that uniformly adapts the impact of each task on the training updates of the agent.

## 7 CONCLUSION

We have theoretically proved the advantage in RL of using a shared representation to learn multiple tasks w.r.t. learning a single task. We have derived our results extending the AVI/API bounds (Farahmand, 2011) to MTRL, leveraging the upper bounds on the approximation error in MTL provided in Maurer et al. (2016). The results of this analysis show that the error propagation during the AVI/API iterations is reduced according to the number of tasks. Then, we proposed a practical way of exploiting this theoretical benefit which consists in an effective way of extracting shared representations of multiple tasks by means of deep neural networks. To empirically show the advantages of our method, we carried out experiments on challenging RL problems with the introduction of multi-task extensions of FQI, DQN, and DDPG based on the neural network structure we proposed. As desired, the favorable empirical results confirm the theoretical benefit we described.

ACKNOWLEDGMENTS

This project has received funding from the European Union's Horizon 2020 research and innovation programme under grant agreement No. #640554 (SKILLS4ROBOTS) and No. #713010 (GOAL-Robots). This project has also been supported by grants from NVIDIA, the NVIDIA DGX Station, and the Intel® AI DevCloud. The authors thank Alberto Maria Metelli, Andrea Tirinzoni and Matteo Papini for their helpful insights during the development of the project.

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

# A PROOFS

## A.1 APPROXIMATED VALUE-ITERATION BOUNDS

*Proof of Theorem 2.* We compute the average expected loss across tasks:

$$\frac{1}{T}\sum_{t=1}^{T}\|Q_t^* - Q_t^{\pi_K}\|_{1,\rho}$$

$$\leq \frac{1}{T}\sum_{t=1}^{T}\frac{2\gamma_t}{(1-\gamma_t)^2}\left[\inf_{r\in[0,1]}C_{\mathrm{VI},\rho,\nu}^{\frac{1}{2}}(K;t,r)\mathcal{E}^{\frac{1}{2}}(\varepsilon_{t,0},\ldots,\varepsilon_{t,K-1};t,r) + \frac{2}{1-\gamma_t}\gamma_t^K R_{\max,t}\right]$$

$$\leq \frac{2\gamma}{(1-\gamma)^2}\frac{1}{T}\sum_{t=1}^{T}\left[\inf_{r\in[0,1]}C_{\mathrm{VI},\rho,\nu}^{\frac{1}{2}}(K;t,r)\mathcal{E}^{\frac{1}{2}}(\varepsilon_{t,0},\ldots,\varepsilon_{t,K-1};t,r) + \frac{2}{1-\gamma_t}\gamma_t^K R_{\max,t}\right]$$

$$\leq \frac{2\gamma}{(1-\gamma)^2}\left[\frac{1}{T}\sum_{t=1}^{T}\left(\inf_{r\in[0,1]}C_{\mathrm{VI},\rho,\nu}^{\frac{1}{2}}(K;t,r)\mathcal{E}^{\frac{1}{2}}(\varepsilon_{t,0},\ldots,\varepsilon_{t,K-1};t,r)\right) + \frac{2}{1-\gamma}\gamma^K R_{\max,\mathrm{avg}}\right]$$

$$\leq \frac{2\gamma}{(1-\gamma)^2}\left[\inf_{r\in[0,1]}\frac{1}{T}\sum_{t=1}^{T}\left(C_{\mathrm{VI},\rho,\nu}^{\frac{1}{2}}(K;t,r)\mathcal{E}^{\frac{1}{2}}(\varepsilon_{t,0},\ldots,\varepsilon_{t,K-1};t,r)\right) + \frac{2}{1-\gamma}\gamma^K R_{\max,\mathrm{avg}}\right]$$

$$\leq \frac{2\gamma}{(1-\gamma)^2}\left[\inf_{r\in[0,1]}C_{\mathrm{VI}}^{\frac{1}{2}}(K;r)\frac{1}{T}\sum_{t=1}^{T}\left(\mathcal{E}^{\frac{1}{2}}(\varepsilon_{t,0},\ldots,\varepsilon_{t,K-1};t,r)\right) + \frac{2}{1-\gamma}\gamma^K R_{\max,\mathrm{avg}}\right] \quad (11)$$

with $\gamma = \max_{t\in\{1,\ldots,T\}}\gamma_t$, $C_{\mathrm{VI}}^{\frac{1}{2}}(K;r) = \max_{t\in\{1,\ldots,T\}}C_{\mathrm{VI},\rho,\nu}^{\frac{1}{2}}(K;t,r)$, and $R_{\max,\mathrm{avg}} = 1/T\sum_{t=1}^{T}R_{\max,t}$.

Considering the term $1/T\sum_{t=1}^{T}\left[\mathcal{E}^{\frac{1}{2}}(\varepsilon_{t,0},\ldots,\varepsilon_{t,K-1};t,r)\right] = 1/T\sum_{t=1}^{T}\left(\sum_{k=0}^{K-1}\alpha_{t,k}^{2r}\varepsilon_{t,k}\right)^{\frac{1}{2}}$ let

$$\alpha_k = \begin{cases} \frac{(1-\gamma)\gamma^{K-k-1}}{1-\gamma^{K+1}} & 0\leq k < K, \\ \frac{(1-\gamma)\gamma^{K}}{1-\gamma^{K+1}} & k = K \end{cases},$$

then we bound

$$\frac{1}{T}\sum_{t=1}^{T}\left(\sum_{k=0}^{K-1}\alpha_{t,k}^{2r}\varepsilon_{t,k}\right)^{\frac{1}{2}} \leq \frac{1}{T}\sum_{t=1}^{T}\left(\sum_{k=0}^{K-1}\alpha_k^{2r}\varepsilon_{t,k}\right)^{\frac{1}{2}}.$$

Using Jensen's inequality:

$$\frac{1}{T}\sum_{t=1}^{T}\left(\sum_{k=0}^{K-1}\alpha_k^{2r}\varepsilon_{t,k}\right)^{\frac{1}{2}} \leq \left(\sum_{k=0}^{K-1}\alpha_k^{2r}\frac{1}{T}\sum_{t=1}^{T}\varepsilon_{t,k}\right)^{\frac{1}{2}}.$$

So, now we can write (11) as

$$\frac{1}{T}\sum_{t=1}^{T}\|Q_t^* - Q_t^{\pi_K}\|_{1,\rho} \leq \frac{2\gamma}{(1-\gamma)^2}\left[\inf_{r\in[0,1]}C_{\mathrm{VI}}^{\frac{1}{2}}(K;r)\mathcal{E}_{\mathrm{avg}}^{\frac{1}{2}}(\varepsilon_{\mathrm{avg},0},\ldots,\varepsilon_{\mathrm{avg},K-1};r)\right.$$

$$\left.+\frac{2}{1-\gamma}\gamma^K R_{\max,\mathrm{avg}}\right],$$

with $\varepsilon_{\mathrm{avg},k} = 1/T\sum_{t=1}^{T}\varepsilon_{t,k}$ and $\mathcal{E}_{\mathrm{avg}}(\varepsilon_{\mathrm{avg},0},\ldots,\varepsilon_{\mathrm{avg},K-1};r) = \sum_{k=0}^{K-1}\alpha_k^{2r}\varepsilon_{\mathrm{avg},k}$.

$\square$

*Proof of Lemma 4.* Let us start from the definition of optimal task-averaged risk:

$$\varepsilon_{\mathrm{avg},k}^* = \frac{1}{T}\sum_{t=1}^{T}\|Q_{t,k+1}^* - \mathcal{T}_t^* Q_{t,k}\|_{\nu}^2,$$

where $Q^*_{t,k}$, with $t \in [1, T]$, are the minimizers of $\varepsilon_{\mathrm{avg},k}$.

Consider the task $\check{t}$ such that

$$\check{t}_k = \arg\max_{t \in \{1,\dots,T\}} \|Q^*_{t,k+1} - \mathcal{T}^*_t Q_{t,k}\|^2_\nu,$$

we can write the following inequality:

$$\sqrt{\varepsilon^*_{\mathrm{avg},k}} \leq \|Q^*_{\check{t}_k,k+1} - \mathcal{T}^*_{\check{t}} Q_{\check{t}_k,k}\|_\nu.$$

By the application of Theorem 5.3 by Farahmand (2011) to the right hand side, and defining $b_{k,i} = \|Q_{\check{t}_k,i+1} - \mathcal{T}^*_{\check{t}} Q_{\check{t}_k,i}\|_\nu$, we obtain:

$$\sqrt{\varepsilon^*_{\mathrm{avg},k}} \leq \|Q^*_{\check{t}_k,k+1} - (\mathcal{T}^*_{\check{t}})^{k+1} Q_{\check{t}_k,0}\|_\nu + \sum_{i=0}^{k-1} (\gamma_{\check{t}_k} C_{\mathrm{AE}}(\nu; \check{t}_k, P))^{i+1} b_{k,k-1-i}.$$

Squaring both sides yields the result:

$$\varepsilon^*_{\mathrm{avg},k} \leq \left( \|Q^*_{\check{t}_k,k+1} - (\mathcal{T}^*_{\check{t}})^{k+1} Q_{\check{t}_k,0}\|_\nu + \sum_{i=0}^{k-1} (\gamma_{\check{t}_k} C_{\mathrm{AE}}(\nu; \check{t}_k, P))^{i+1} b_{k,k-1-i} \right)^2.$$

$\square$

## A.2 APPROXIMATED POLICY-ITERATION BOUNDS

We start by considering the bound for the API framework:

**Theorem 5.** *(Theorem 3.2 of Farahmand (2011)) Let K be a positive integer, and $Q_{max} \leq \frac{R_{max}}{1-\gamma}$. Then for any sequence $(Q_k)_{k=0}^{K-1} \subset B(\mathcal{S} \times \mathcal{A}, Q_{max})$ and the corresponding sequence $(\varepsilon_k)_{k=0}^{K-1}$, where $\varepsilon_k = \|Q_k - Q^{\pi_k}\|^2_\nu$, we have:*

$$\|Q^* - Q^{\pi_K}\|_{1,\rho} \leq \frac{2\gamma}{(1-\gamma)^2} \left[ \inf_{r \in [0,1]} C^{\frac{1}{2}}_{PI,\rho,\nu}(K; r) \mathcal{E}^{\frac{1}{2}}(\varepsilon_0, \dots, \varepsilon_{K-1}; r) + \gamma^{K-1} R_{max} \right], \quad (12)$$

*where*

$$C_{PI,\rho,\nu}(K; r) =$$

$$\left( \frac{1-\gamma}{2} \right)^2 \sup_{\pi'_0,\dots,\pi'_K} \sum_{k=0}^{K-1} a_k^{2(1-r)} \left( \sum_{m \geq 0} \gamma^m c_{PI_1,\rho,\nu}(K-k-1, m+1; \pi'_{k+1}) + \right.$$

$$\left. \sum_{m \geq 1} \gamma^m c_{PI_2,\rho,\nu}(K-k-1, m; \pi'_{k+1}, \pi'_k) + c_{PI_3,\rho,\nu} \right)^2;$$

$$(13)$$

*with $\mathcal{E}(\varepsilon_0, \dots, \varepsilon_{K-1}; r) = \sum_{k=0}^{K-1} \alpha_k^{2r} \varepsilon_k$, the three coefficients $c_{PI_1,\rho,\nu}$, $c_{PI_2,\rho,\nu}$, $c_{PI_3,\rho,\nu}$, the distributions $\rho$ and $\nu$, and the series $\alpha_k$ are defined as in Farahmand (2011).*

From Theorem 5, by computing the average loss across tasks and pushing inside the average using Jensen's inequality, we derive the API bounds averaged on multiple tasks.

**Theorem 6.** *Let K be a positive integer, and $Q_{max} \leq \frac{R_{max}}{1-\gamma}$. Then for any sequence $(Q_k)_{k=0}^{K-1} \subset B(\mathcal{S} \times \mathcal{A}, Q_{max})$ and the corresponding sequence $(\varepsilon_{avg,k})_{k=0}^{K-1}$, where $\varepsilon_{avg,k} = \frac{1}{T} \sum_{t=1}^{T} \|Q_{t,k} - Q^{\pi_k}_t\|^2_\nu$, we have:*

$$\frac{1}{T} \sum_{t=1}^{T} \|Q^*_t - Q^{\pi_K}_t\|_{1,\rho} \leq \frac{2\gamma}{(1-\gamma)^2} \left[ \inf_{r \in [0,1]} C^{\frac{1}{2}}_{PI}(K; r) \mathcal{E}^{\frac{1}{2}}_{avg}(\varepsilon_{avg,0}, \dots, \varepsilon_{avg,K-1}; r) \right.$$

$$\left. + \gamma^{K-1} R_{max,avg} \right], \quad (14)$$

*with* $\mathcal{E}_{avg} = \sum_{k=0}^{K-1} \alpha_k^{2r} \varepsilon_{avg,k}$, $\gamma = \max_{t \in \{1,\dots,T\}} \gamma_t$, $C_{PI}^{\frac{1}{2}}(K;r) = \max_{t \in \{1,\dots,T\}} C_{PI,\rho,\nu}^{\frac{1}{2}}(K;t,r)$, $R_{max,avg} =$
$\frac{1}{T} \sum_{t=1}^{T} R_{max,t}$ *and* $\alpha_k = \begin{cases} \frac{(1-\gamma)\gamma^{K-k-1}}{1-\gamma^{K+1}} & 0 \le k < K, \\ \frac{(1-\gamma)\gamma^K}{1-\gamma^{K+1}} & k = K \end{cases}$.

*Proof of Theorem 6.* The proof is very similar to the one for AVI. We compute the average expected loss across tasks:

$$\frac{1}{T} \sum_{t=1}^{T} \|Q_t^* - Q_t^{\pi_K}\|_{1,\rho}$$

$$\le \frac{1}{T} \sum_{t=1}^{T} \frac{2\gamma_t}{(1-\gamma_t)^2} \left[ \inf_{r \in [0,1]} C_{PI,\rho,\nu}^{\frac{1}{2}}(K;t,r) \mathcal{E}^{\frac{1}{2}}(\varepsilon_{t,0},\dots,\varepsilon_{t,K-1};t,r) + \gamma_t^{K-1} R_{max,t} \right]$$

$$\le \frac{2\gamma}{(1-\gamma)^2} \frac{1}{T} \sum_{t=1}^{T} \left[ \inf_{r \in [0,1]} C_{PI,\rho,\nu}^{\frac{1}{2}}(K;t,r) \mathcal{E}^{\frac{1}{2}}(\varepsilon_{t,0},\dots,\varepsilon_{t,K-1};t,r) + \gamma_t^{K-1} R_{max,t} \right]$$

$$\le \frac{2\gamma}{(1-\gamma)^2} \left[ \frac{1}{T} \sum_{t=1}^{T} \left( \inf_{r \in [0,1]} C_{PI,\rho,\nu}^{\frac{1}{2}}(K;t,r) \mathcal{E}^{\frac{1}{2}}(\varepsilon_{t,0},\dots,\varepsilon_{t,K-1};t,r) \right) + \gamma^{K-1} R_{max,avg} \right]$$

$$\le \frac{2\gamma}{(1-\gamma)^2} \left[ \inf_{r \in [0,1]} \frac{1}{T} \sum_{t=1}^{T} \left( C_{PI,\rho,\nu}^{\frac{1}{2}}(K;t,r) \mathcal{E}^{\frac{1}{2}}(\varepsilon_{t,0},\dots,\varepsilon_{t,K-1};t,r) \right) + \gamma^{K-1} R_{max,avg} \right]$$

$$\le \frac{2\gamma}{(1-\gamma)^2} \left[ \inf_{r \in [0,1]} C_{PI}^{\frac{1}{2}}(K;r) \frac{1}{T} \sum_{t=1}^{T} \left( \mathcal{E}^{\frac{1}{2}}(\varepsilon_{t,0},\dots,\varepsilon_{t,K-1};t,r) \right) + \gamma^{K-1} R_{max,avg} \right]. \quad (15)$$

Using Jensen's inequality as in the AVI scenario, we can write (15) as:

$$\frac{1}{T} \sum_{t=1}^{T} \|Q_t^* - Q_t^{\pi_K}\|_{1,\rho} \le \frac{2\gamma}{(1-\gamma)^2} \left[ \inf_{r \in [0,1]} C_{PI}^{\frac{1}{2}}(K;r) \mathcal{E}_{avg}^{\frac{1}{2}}(\varepsilon_{avg,0},\dots,\varepsilon_{avg,K-1};r) \right.$$
$$\left. + \gamma^{K-1} R_{max,avg} \right], \quad (16)$$

with $\varepsilon_{avg,k} = 1/T \sum_{t=1}^{T} \varepsilon_{t,k}$ and $\mathcal{E}_{avg}(\varepsilon_{avg,0},\dots,\varepsilon_{avg,K-1};r) = \sum_{k=0}^{K-1} \alpha_k^{2r} \varepsilon_{avg,k}$. $\qquad\square$

### A.3 Approximation bounds

*Proof of Theorem 3.* Let $w_1^*,\dots,w_T^*$, $h^*$ and $f_1^*,\dots,f_T^*$ be the minimizers of $\varepsilon_{avg}^*$, then:

$$\varepsilon_{avg}(\hat{\mathbf{w}},\hat{h},\hat{\mathbf{f}}) - \varepsilon_{avg}^* = \underbrace{\left( \varepsilon_{avg}(\hat{\mathbf{w}},\hat{h},\hat{\mathbf{f}}) - \frac{1}{nT} \sum_{ti} \ell(\hat{f}_t(\hat{h}(\hat{w}_t(X_{ti}))), Y_{ti}) \right)}_{A}$$

$$+ \underbrace{\left( \frac{1}{nT} \sum_{ti} \ell(\hat{f}_t(\hat{h}(\hat{w}_t(X_{ti}))), Y_{ti}) - \frac{1}{nT} \sum_{ti} \ell(f_t^*(h^*(w_t^*(X_{ti}))), Y_{ti}) \right)}_{B}$$

$$+ \underbrace{\left( \frac{1}{nT} \sum_{ti} \ell(f_t^*(h^*(w_t^*(X_{ti}))), Y_{ti}) - \varepsilon_{avg}^* \right)}_{C}. \quad (17)$$

We proceed to bound the three components individually:

- $C$ can be bounded using Hoeffding's inequality, with probability $1 - \delta/2$ by $\sqrt{\ln(2/\delta)/(2nT)}$, as it contains only $nT$ random variables bounded in the interval $[0,1]$;

- $B$ can be bounded by $0$, by definition of $\hat{\mathbf{w}}$, $\hat{h}$ and $\hat{\mathbf{f}}$, as they are the minimizers of Equation (3);

- the bounding of $A$ is less straightforward and is described in the following.

We define the following auxiliary function spaces:

- $\mathcal{W}' = \{x \in \mathcal{X} \to (w_t(x_{ti})) : (w_1, \ldots, w_T) \in \mathcal{W}^T\}$,

- $\mathcal{F}' = \{y \in \mathbb{R}^{KTn} \to (f_t(y_{ti})) : (f_1, \ldots, f_T) \in \mathcal{F}^T\}$,

and the following auxiliary sets:

- $S = \{(\ell(f_t(h(w_t(X_{ti}))), Y_{ti})) : f \in \mathcal{F}^T, h \in \mathcal{H}, w \in \mathcal{W}^T\} \subseteq \mathbb{R}^{Tn}$,

- $S' = \mathcal{F}'(\mathcal{H}(\mathcal{W}'(\bar{\mathbf{X}}))) = \{(f_t(h(w_t(X_{ti})))) : f \in \mathcal{F}^T, h \in \mathcal{H}, w \in \mathcal{W}^T\} \subseteq \mathbb{R}^{Tn}$,

- $S'' = \mathcal{H}(\mathcal{W}'(\bar{\mathbf{X}})) = \{(h(w_t(X_{ti}))) : h \in \mathcal{H}, w \in \mathcal{W}^T\} \subseteq \mathbb{R}^{KTn}$,

which will be useful in our proof.

Using Theorem 9 by Maurer et al. (2016), we can write:

$$\varepsilon_{\text{avg}}(\hat{\mathbf{w}}, \hat{h}, \hat{\mathbf{f}}) - \frac{1}{nT}\sum_{ti}\ell(\hat{f}_t(\hat{h}(\hat{w}_t(X_{ti}))), Y_{ti})$$

$$\leq \sup_{\mathbf{w}\in\mathcal{W}^T, h\in\mathcal{H}, \mathbf{f}\in\mathcal{F}^T}\left(\varepsilon_{\text{avg}}(\mathbf{w}, h, \mathbf{f}) - \frac{1}{nT}\sum_{ti}\ell(f_t(h(w_t(X_{ti}))), Y_{ti})\right)$$

$$\leq \frac{\sqrt{2\pi}G(S)}{nT} + \sqrt{\frac{9\ln(\frac{2}{\delta})}{2nT}}, \tag{18}$$

then by Lipschitz property of the loss function $\ell$ and the contraction lemma Corollary 11 Maurer et al. (2016): $G(S) \leq G(S')$. By Theorem 12 by Maurer et al. (2016), for universal constants $c'_1$ and $c'_2$:

$$G(S') \leq c'_1 L(\mathcal{F}')G(S'') + c'_2 D(S'')O(\mathcal{F}') + \min_{y\in Y} G(\mathcal{F}(y)), \tag{19}$$

where $L(\mathcal{F}')$ is the largest value for the Lipschitz constants in the function space $\mathcal{F}'$, and $D(S'')$ is the Euclidean diameter of the set $S''$.

Using Theorem 12 by Maurer et al. (2016) again, for universal constants $c''_1$ and $c''_2$:

$$G(S'') \leq c''_1 L(\mathcal{H})G(\mathcal{W}'(\bar{\mathbf{X}})) + c''_2 D(\mathcal{W}'(\bar{\mathbf{X}}))O(\mathcal{H}) + \min_{p\in P} G(\mathcal{H}(p)). \tag{20}$$

Putting (19) and (20) together:

$$G(S') \leq c'_1 L(\mathcal{F}')\left(c''_1 L(\mathcal{H})G(\mathcal{W}'(\bar{\mathbf{X}})) + c''_2 D(\mathcal{W}'(\bar{\mathbf{X}}))O(\mathcal{H}) + \min_{p\in P} G(\mathcal{H}(p))\right)$$

$$+ c'_2 D(S'')O(\mathcal{F}') + \min_{y\in Y} G(\mathcal{F}(y))$$

$$= c'_1 c''_1 L(\mathcal{F}')L(\mathcal{H})G(\mathcal{W}'(\bar{\mathbf{X}})) + c'_1 c''_2 L(\mathcal{F}')D(\mathcal{W}'(\bar{\mathbf{X}}))O(\mathcal{H}) + c'_1 L(\mathcal{F}')\min_{p\in P} G(\mathcal{H}(p))$$

$$+ c'_2 D(S'')O(\mathcal{F}') + \min_{y\in Y} G(\mathcal{F}(y)). \tag{21}$$

At this point, we have to bound the individual terms in the right hand side of (21), following the same procedure proposed by Maurer et al. (2016).

Firstly, to bound $L(\mathcal{F}')$, let $y, y' \in \mathbb{R}^{KTn}$, where $y = (y_{ti})$ with $y_{ti} \in \mathbb{R}^K$ and $y' = (y'_{ti})$ with $y'_{ti} \in \mathbb{R}^K$. We can write the following:

$$
\begin{aligned}
\|f(y) - f(y')\|^2 &= \sum_{ti} \left( f_t(y_{ti}) - f_t(y'_{ti}) \right)^2 \\
&\leq L(\mathcal{F})^2 \sum_{ti} \|y_{ti} - y'_{ti}\|^2 \\
&= L(\mathcal{F})^2 \|y - y'\|^2,
\end{aligned}
\tag{22}
$$

whence $L(\mathcal{F}') \leq L(\mathcal{F})$.

Then, we bound:

$$
\begin{aligned}
G(\mathcal{W}'(\bar{\mathbf{X}})) = \mathbb{E}\left[ \sup_{\mathbf{w} \in \mathcal{W}^T} \sum_{kti} \gamma_{kti} w_{tk}(X_{ti}) \Big| X_{ti} \right] &\leq \sum_t \sup_{l \in \{1,\dots,T\}} \mathbb{E}\left[ \sup_{w \in \mathcal{W}} \sum_{ki} \gamma_{kli} w_k(X_{li}) \Big| X_{li} \right] \\
&= T \sup_{l \in \{1,\dots,T\}} G(\mathcal{W}(\mathbf{X}_l)).
\end{aligned}
\tag{23}
$$

Then, since it is possible to bound the Euclidean diameter using the norm of the supremum value in the set, we bound $D(S'') \leq 2 \sup_{h,\mathbf{w}} \|h(\mathbf{w}(\bar{\mathbf{X}}))\|$ and $D(\mathcal{W}'(\bar{\mathbf{X}})) \leq 2 \sup_{\mathbf{w} \in \mathcal{W}^T} \|\mathbf{w}(\bar{\mathbf{X}})\|$.

Also, we bound $O(\mathcal{F}')$:

$$
\begin{aligned}
\mathbb{E}\left[ \sup_{g \in \mathcal{F}'} \langle \boldsymbol{\gamma}, g(y) - g(y') \rangle \right] &= \mathbb{E}\left[ \sup_{\boldsymbol{f} \in \mathcal{F}^T} \sum_{ti} \gamma_{ti} \left( f_t(y_{ti}) - f_t(y'_{ti}) \right) \right] \\
&= \sum_t \mathbb{E}\left[ \sup_{f \in \mathcal{F}} \sum_i \gamma_i \left( f(y_{ti}) - f(y'_{ti}) \right) \right] \\
&\leq \sqrt{T} \left( \sum_t \mathbb{E}\left[ \sup_{f \in \mathcal{F}} \sum_i \gamma_i \left( f(y_{ti}) - f(y'_{ti}) \right) \right]^2 \right)^{\frac{1}{2}} \\
&\leq \sqrt{T} \left( \sum_t O(\mathcal{F})^2 \sum_i \|y_{ti} - y'_{ti}\|^2 \right)^{\frac{1}{2}} \\
&= \sqrt{T} O(\mathcal{F}) \|y - y'\|,
\end{aligned}
\tag{24}
$$

whence $O(\mathcal{F}') \leq \sqrt{T} O(\mathcal{F})$.

To minimize the last term, it is possible to choose $y_0 = 0$, as $f(0) = 0, \forall f \in \mathcal{F}$, resulting in $\min_{y \in Y} G(\mathcal{F}(y)) = G(\mathcal{F}(0)) = 0$.

Then, substituting in (21), and recalling that $G(S) \leq G(S')$:

$$
\begin{aligned}
G(S) \leq{}& c'_1 c''_1 L(\mathcal{F}) L(\mathcal{H}) T \sup_{l \in \{1,\dots,T\}} G(\mathcal{W}(\mathbf{X}_l)) + 2 c'_1 c''_2 L(\mathcal{F}) \sup_{\mathbf{w} \in \mathcal{W}^T} \|\mathbf{w}(\bar{\mathbf{X}})\| O(\mathcal{H}) \\
&+ c'_1 L(\mathcal{F}) \min_{p \in P} G(\mathcal{H}(p)) + 2 c'_2 \sup_{h,\mathbf{w}} \|h(\mathbf{w}(\bar{\mathbf{X}}))\| \sqrt{T} O(\mathcal{F}).
\end{aligned}
\tag{25}
$$

Now, the first term $A$ of (17) can be bounded substituting (25) in (18):

$$\varepsilon_{\text{avg}}(\hat{\mathbf{w}}, \hat{h}, \hat{\mathbf{f}}) - \frac{1}{nT} \sum_{ti} \ell(\hat{f}_t(\hat{h}(\hat{w}_t(X_{ti}))), Y_{ti})$$

$$\leq \frac{\sqrt{2\pi}}{nT} \left( c_1' c_1'' L(\mathcal{F}) L(\mathcal{H}) T \sup_{l \in \{1,\ldots,T\}} G(\mathcal{W}(\mathbf{X}_l)) + 2c_1' c_2'' L(\mathcal{F}) \sup_{\mathbf{w} \in \mathcal{W}^T} \|\mathbf{w}(\bar{\mathbf{X}})\| O(\mathcal{H}) \right.$$

$$\left. + c_1' L(\mathcal{F}) \min_{p \in P} G(\mathcal{H}(p)) + 2c_2' \sup_{h,\mathbf{w}} \|h(\mathbf{w}(\bar{\mathbf{X}}))\| \sqrt{T} O(\mathcal{F}) \right) + \sqrt{\frac{9 \ln(\frac{2}{\delta})}{2nT}}$$

$$= c_1 \frac{L(\mathcal{F}) L(\mathcal{H}) \sup_{l \in \{1,\ldots,T\}} G(\mathcal{W}(\mathbf{X}_l))}{n} + c_2 \frac{\sup_{\mathbf{w}} \|\mathbf{w}(\bar{\mathbf{X}})\| L(\mathcal{F}) O(\mathcal{H})}{nT}$$

$$+ c_3 \frac{L(\mathcal{F}) \min_{p \in P} G(\mathcal{H}(p))}{nT} + c_4 \frac{\sup_{h,\mathbf{w}} \|h(\mathbf{w}(\bar{\mathbf{X}}))\| O(\mathcal{F})}{n\sqrt{T}} + \sqrt{\frac{9 \ln(\frac{2}{\delta})}{2nT}}.$$

A union bound between $A$, $B$ and $C$ of (17) completes the proof:

$$\varepsilon_{\text{avg}}(\hat{\mathbf{w}}, \hat{h}, \hat{\mathbf{f}}) - \varepsilon_{\text{avg}}^* \leq c_1 \frac{L(\mathcal{F}) L(\mathcal{H}) \sup_{l \in \{1,\ldots,T\}} G(\mathcal{W}(\mathbf{X}_l))}{n}$$

$$+ c_2 \frac{\sup_{\mathbf{w}} \|\mathbf{w}(\bar{\mathbf{X}})\| L(\mathcal{F}) O(\mathcal{H})}{nT}$$

$$+ c_3 \frac{L(\mathcal{F}) \min_{p \in P} G(\mathcal{H}(p))}{nT}$$

$$+ c_4 \frac{\sup_{h,\mathbf{w}} \|h(\mathbf{w}(\bar{\mathbf{X}}))\| O(\mathcal{F})}{n\sqrt{T}}$$

$$+ \sqrt{\frac{8 \ln(\frac{3}{\delta})}{nT}}.$$

$$\square$$

# B ADDITIONAL DETAILS OF EMPIRICAL EVALUATION

## B.1 MULTI FITTED $Q$-ITERATION

We consider *Car-On-Hill* problem with discount factor $0.95$ and horizon $100$. Running Adam optimizer with learning rate $0.001$ and using a mean squared loss, we train a neural network composed of $2$ shared layers of $30$ neurons each, with sigmoidal activation function, as described in Riedmiller (2005). We select $8$ tasks for the problem changing the mass of the car $m$ and the value of the discrete actions $a$ (Table 1). Figure 1(b) is computed considering the first four tasks, while Figure 1(c) considers task 1 in the result with 1 task, tasks 1 and 2 for the result with 2 tasks, tasks 1, 2, 3, and 4 for the result with 4 tasks, and all the tasks for the result with 8 tasks. To run FQI and MFQI, for each task we collect transitions running an extra-tree trained following the procedure and setting in Ernst et al. (2005), using an $\epsilon$-greedy policy with $\epsilon = 0.1$, to obtain a small, but representative dataset. The optimal $Q$-function for each task is computed by tree-search[3] for $100$ states uniformly picked from the state space, and the $2$ discrete actions, for a total of $200$ state-action tuples.

## B.2 MULTI DEEP $Q$-NETWORK

The five problems we consider for this experiment are: *Cart-Pole*, *Acrobot*, *Mountain-Car*, *Car-On-Hill*, and *Inverted-Pendulum*[4]. The discount factors are respectively $0.99$, $0.99$, $0.99$, $0.95$, and $0.95$. The horizons are respectively $500$, $1,000$, $1,000$, $100$, and $3,000$. The network we use consists of $80$ ReLu units for each $w_t, t \in \{1, \ldots, T\}$ block, with $T = 5$. Then, the shared block $h$ consists of one

---

[3]We follow the method described in Ernst et al. (2005).

[4]The IDs of the problems in the OpenAI Gym library are: *CartPole-v0*, *Acrobot-v1*, and *MountainCar-v0*.

| Task | Mass | Action set |
|------|------|------------|
| 1 | 1.0 | $\{-4.0; 4.0\}$ |
| 2 | 0.8 | $\{-4.0; 4.0\}$ |
| 3 | 1.0 | $\{-4.5; 4.5\}$ |
| 4 | 1.2 | $\{-4.5; 4.5\}$ |
| 5 | 1.0 | $\{-4.125; 4.125\}$ |
| 6 | 1.0 | $\{-4.25; 4.25\}$ |
| 7 | 0.8 | $\{-4.375; 4.375\}$ |
| 8 | 0.85 | $\{-4.0; 4.0\}$ |

Table 1: Different values of the mass of the car and available actions chosen for the Car-On-Hill tasks in the MFQI empirical evaluation.

layer with 80 ReLu units and another one with 80 sigmoid units. Eventually, each $f_t$ has a number of linear units equal to the number of discrete actions $a_i^{(t)}, i \in \{1, \ldots, \#\mathcal{A}^{(t)}\}$ of task $\mu_t$ which outputs the action-value $Q_t(s, a_i^{(t)}) = y_t(s, a_i^{(t)}) = f_t(h(w_t(s)), a_i^{(t)}), \forall s \in \mathcal{S}^{(t)}$. The use of sigmoid units in the second layer of $h$ is due to our choice to extract meaningful shared features bounded between 0 and 1 to be used as input of the last linear layer, as in most RL approaches. In practice, we have also found that sigmoid units help to reduce task interference in multi-task networks, where instead the linear response of ReLu units cause a problematic increase in the feature values. Furthermore, the use of a bounded feature space reduces the $\sup_{h,\mathbf{w}} \|h(\mathbf{w}(\bar{\mathbf{X}}))\|$ term in the upper bound of Theorem 3, corresponding to the upper bound of the diameter of the feature space, as shown in Appendix A. The initial replay memory size for each task is 100 and the maximum size is $5,000$. We use Huber loss with Adam optimizer using learning rate $10^{-3}$ and batch size of 100 samples for each task. The target network is updated every 100 steps. The exploration is $\varepsilon$-greedy with $\varepsilon$ linearly decaying from 1 to 0.01 in the first $5,000$ steps.

## B.3 MULTI DEEP DETERMINISTIC POLICY GRADIENT

The two set of problems we consider for this experiment are: one including *Inverted-Pendulum*, *Inverted-Double-Pendulum*, and *Inverted-Pendulum-Swingup*, and another one including *Hopper-Stand*, *Walker-Walk*, and *Half-Cheetah-Run*[5]. The discount factors are 0.99 and the horizons are $1,000$ for all problems. The actor network is composed of 600 ReLu units for each $w_t, t \in \{1, \ldots, T\}$ block, with $T = 3$. The shared block $h$ has 500 units with ReLu activation function as for MDQN. Finally, each $f_t$ has a number of *tanh* units equal to the number of dimensions of the continuous actions $a^{(t)} \in \mathcal{A}^{(t)}$ of task $\mu_t$ which outputs the policy $\pi_t(s) = y_t(s) = f_t(h(w_t(s))), \forall s \in \mathcal{S}^{(t)}$. On the other hand, the critic network consists of the same $w_t$ units of the actor, except for the use of sigmoidal units in the $h$ layer, as in MDQN. In addition to this, the actions $a^{(t)}$ are given as input to $h$. Finally, each $f_t$ has a single linear unit $Q_t(s, a^{(t)}) = y_t(s, a^{(t)}) = f_t(h(w_t(s), a^{(t)})), \forall s \in \mathcal{S}^{(t)}$. The initial replay memory size for each task is 64 and the maximum size is $50,000$. We use Huber loss to update the critic network and the policy gradient to update the actor network. In both cases the optimization is performed with Adam optimizer and batch size of 64 samples for each task. The learning rate of the actor is $10^{-4}$ and the learning rate of the critic is $10^{-3}$. Moreover, we apply $\ell_2$-penalization to the critic network using a regularization coefficient of 0.01. The target networks are updated with soft-updates using $\tau = 10^{-3}$. The exploration is performed using the action computed by the actor network adding a noise generated with an Ornstein-Uhlenbeck process with $\theta = 0.15$ and $\sigma = 0.2$. Note that most of these values are taken from the original DDPG paper Lillicrap et al. (2015), which optimizes them for the single-task scenario.

---

[5]The IDs of the problems in the pybullet library are: *InvertedPendulumBulletEnv-v0*, *InvertedDoublePendulumBulletEnv-v0*, and *InvertedPendulumSwingupBulletEnv-v0*. The names of the domain and the task of the problems in the DeepMind Control Suite are: *hopper-stand*, *walker-walk*, and *cheetah-run*.

