# OpenReview forum: "Sharing Knowledge in Multi-Task Deep Reinforcement Learning"
_ICLR.cc/2020/Conference — Accept (Poster)_

### Official Review · AnonReviewer2 · 2019-10-22
**Official Blind Review #2**

**Rating:** 6

**Review:**

The submission derives bounds for approximate value and policy iteration for the multitask case in reinforcement learning. In addition, two common RL algorithms are adapted to demonstrate benefits of multitask RL given related tasks.

The paper is mostly well written but sometimes introduces potentially unnecessarily complex mathematical notation including missing and dual definitions which slow down the reader. Examples of missing definitions are given by K an n on top of page 3.

Generelly, the paper is quite self consist thanks to minor changes in existing algorithms to show experimental results for the theoretical insights, but instead of requiring the reader to find other papers it would be better to define all terminology in the appendix (see equation 6).

The main contribution of the submission is the extension of an existing bound to the multitask case as the architectures are common across existing work. The extension is relevant given growing interest in meta and multitask RL but the changes in comparison to the single task case are minor and the experiment’s value purely lies in supporting this extension.


**Experience Assessment:**

I have published one or two papers in this area.

**Review Assessment: Checking Correctness Of Derivations And Theory:**

I assessed the sensibility of the derivations and theory.

**Review Assessment: Checking Correctness Of Experiments:**

I assessed the sensibility of the experiments.

**Review Assessment: Thoroughness In Paper Reading:**

I read the paper at least twice and used my best judgement in assessing the paper.

---

> ### Author Response · Authors · 2019-11-08
> **Response to Official Blind Review #2**
>
> We thank the reviewer for reading the paper and his/her positive feedback.
>
> We agree about the missing definitions of $n$ and $K$, and we have added their definition in the already uploaded revision of the paper. We are not sure about the dual definitions reported by the reviewer, so we will be glad if the reviewer can clarify this.
>
> The main contribution of the paper consists of the theoretical analysis of the benefit of sharing representation between tasks in RL. Despite several promising empirical results in the literature, to the best of our knowledge, we are the first to theoretically prove this benefit. The purpose of the three algorithms we introduced is twofold:
> - proving the empirical evidence of the consequence of the bound in Theorem 2;
> - showing the benefit in terms of performance in challenging RL problems.
> In the first experiments, we consider the car-on-hill MDP, which despite being a non-trivial RL problem, still allows computing the optimal $Q$-function. Theorem 2 bounds the norm-1 of the difference between the optimal $Q$-function and the $Q$-function at each update of the policy. The consequence is that the approximation of the $Q$-function converges faster to the optimal $Q$-function for an increasing number of tasks. The results in Figure 1(b) and Figure 1(c) provide the empirical evidence of this.
>
> In the subsequent experiments, we only assess the performance improvements, since it is not feasible to compute the optimal $Q$-function of such complex problems. In all these experiments, the performance of the multi-task network is always better or equal than the single-task. Note that the multi-task network is always using fewer parameters than the overall single-task networks, so even achieving the same performance is a remarkable result of our approach.

---

### Official Review · AnonReviewer1 · 2019-10-22
**Official Blind Review #1**

**Rating:** 6

**Review:**

This paper provides a theoretical justification for the benefit of multi-task deep RL (MTRL) with shared representations. By extending prior work (Farahmand (2011) and Maurer et al. (2016)), the authors demonstrate that the bound of MTRL can be improved if the cost of learning the shared representation at each AVI iteration can be reduced, which is mitigated as we increase the number of tasks. The author also empirically verify their theoretical results in a tabular Q-Fitted Iteration domain and also in challenging RL domains such as Mujoco. The results show that MTRL with shared representation can outperform their single task counterparts to some degree.

Overall this paper adapts the theory shown in Farahmand (2011) and Maurer et al. (2016) to the setting of MTRL and demonstrates the effectiveness of using shared layers, which seems intuitive. While the theory seems a bit incremental, it’s the first paper that theoretically validates the benefits of sharing knowledge, which is a contribution to the MTRL field. I would recommend a weak accept, though I have a few concerns on experimental results, and hope that the authors can clarify them during rebuttal.

Specifically, as the authors have noted, there is a wide range of prior works [1,2,3] that have empirically demonstrated the effectiveness of utilizing shared representations in MTRL. While the authors claim that the goal of the experiments is to show that MTRL with shared layers can outperform its sing task counterparts and thus they ignore other MTRL approaches. I believe that is not the main argument of the paper. The authors should provide empirical evidence on the claim that with an increasing number of tasks in MTRL, the error bound should improve and the performance of MTRL should also boost. Besides, I find the comparison where single-task training is initialized with shared representation a bit confusing. Training would definitely be improved when it’s initialized with some related pretrained features. Maybe the authors should compare this to some other methods such as initializing with single-task representation or even representation learned from training different tasks.

[1] M. Hessel, H. Soyer, L. Espeholt, W. Czarnecki,S. Schmitt, and H. van Hasselt. Multi-task deep reinforcement learning with popart.arXiv preprintarXiv:1809.04474, 2018.
[2] Teh, Y.W., Bapst, V., Czarnecki, W.M., Quan, J., Kirkpatrick, J., Hadsell, R.,Heess, N., Pascanu, R.: Distral: Robust multitask reinforcement learning. In: Ad-vances in Neural Information Processing Systems 30: Annual Conference on Neu-ral Information Processing Systems 2017 (2017)
[3] Wulfmeier, M., Abdolmaleki, A., Hafner, R., Springenberg, J. T., Neunert, M., Hertweck, T., ... & Riedmiller, M. (2019). Regularized Hierarchical Policies for Compositional Transfer in Robotics. arXiv preprint arXiv:1906.11228.

**Experience Assessment:**

I have published one or two papers in this area.

**Review Assessment: Checking Correctness Of Derivations And Theory:**

I assessed the sensibility of the derivations and theory.

**Review Assessment: Checking Correctness Of Experiments:**

I assessed the sensibility of the experiments.

**Review Assessment: Thoroughness In Paper Reading:**

I read the paper at least twice and used my best judgement in assessing the paper.

---

> ### Author Response · Authors · 2019-11-08
> **Response to Official Blind Review #1**
>
> We thank the reviewer for reading the paper and the positive comments about it. In the following, we aim to clarify the reported doubts.
>
> We confirm the opinion of the reviewer that this paper aims at providing theoretical guarantees on the intuitive benefit of sharing representation of multiple tasks in Deep RL. Theorem 3 derives the upper bound of the approximation error $\varepsilon_{\text{avg},k}^*$ averaged over multiple tasks, and Theorem 2 derives the first multi-task AVI bound in literature. Since the bound in Theorem 2 contains the task-averaged approximation error $\varepsilon_{\text{avg},k}^*$, which is bounded by the upper bound in Theorem 3 that decreases for an increasing number of tasks, the bound in Theorem 2 shows the benefit of multi-task RL w.r.t. learning a single task. More in detail, Theorem 2 proves that learning multiple tasks together helps to converge to the optimal $Q$-function faster than the single-task scenario.
>
> Regarding the experiments, we firstly address this doubt of the reviewer: “I believe that is not the main argument of the paper. The authors should provide empirical evidence on the claim that with an increasing number of tasks in MTRL, the error bound should improve and the performance of MTRL should also boost.” Since this is definitely something useful, we provide this analysis in Figure 1(b) and Figure 1(c). Theorem 2 bounds the norm-1 of the difference between the optimal $Q$-function and the $Q$-function at each update of the policy; thus, we want to measure the progress of this measure during learning in single-task and multi-task scenarios. Note that, except for very easy tabular problems, it is not easy to compute the optimal $Q$-function in RL. Nevertheless, we choose the car-on-hill MDP which is a not trivial problem that still allows us to compute the optimal $Q$-function. We solve it using Neural Fitted $Q$-Iteration, based on a neural network built accordingly to our proposed architecture. Since a neural network is a parametric regressor, we are not sure about what the reviewer means when it refers to “tabular” Fitted $Q$-Iteration. In the left plot in Figure 1(b), we consider four different tasks in the car-on-hill MDP. We solve each of these tasks with a single-task network, and all of them together in our multi-task network. Then, we show the progress of the averaged norm-1s of each single task network, and the norm-1 of the multi-task network. The plot shows how the multi-task network is able to get closer to the optimal $Q$-function w.r.t. the single task network. Then, Figure 1(c) provides evidence of the benefit of increasing the number of tasks. Note how the approximation of the optimal $Q$-function gets progressively better and more stable for an increasing number of tasks. Moreover, the right plot in Figure 1(b) also shows the benefit of multi-task in terms of performance. We think this experiment shows what the reviewer is asking, and we wonder if this is not clear in the paper. Regarding the initialization of the single-task network, we emphasize that the shared representation is learned on all the other tasks, excluding the one used for training the single-task network. This experiment has the purpose of providing another empirical evidence of the meaningfulness of the extracted shared features. We hope to have clarified the reviewer’s doubts; otherwise, we will be glad to resolve further concerns.
>
> We noted that the last reference provided by the reviewer is not included in our paper; we thank the reviewer for the additional reference and we have added it in the revised version of the paper.

---

### Official Review · AnonReviewer3 · 2019-10-23
**Official Blind Review #3**

**Rating:** 6

**Review:**

The paper attempts to give theoretical support for using shared representations among multiple tasks.  The architecture has already been proposed in another paper.  The main contribution of the paper is the theory that it claims to support this architecture.  However, I am dubious that the architecture achieves the claimed bound.  One, I do not see how the analysis of this paper is connected to this specific architecture.  Two, the analysis follows from an existing paper by Farahmand (2011), which has already established a bound on the difference Q* - Q^{\pi K}.  This paper considers the same difference, separately for each task, so that the same bound from Farahmand (2011) can be trivially used.  The shared layer h would affect the Q-values, but considering only the difference Q_t* - Q_t{\pi K} abstracts away how sharing representations is helpful.  Simply averaging the norm of the difference between Q_t* - Q_t{\pi K} as if they are independent ignores the fact that the Q_t’s are all dependent due to the shared layer h.  Also, I do not think that the proof of Theorem 2 and Theorem 6 can use the bound of Farahmand (2011).   Because the definition of approximation error epsilon_k in Farahmand (2011) is very different from epsilon_{avg,k} used in this paper, this warrants the analysis to start from the beginning, deriving the bounds from relating Q_t* - Q_t^{\pi K} to (\epsilon_{avg, k})_{k=0}^{K-1}.  The experiments compare known algorithms, and I am unsure how they support the theoretical bounds.

Other comments:
Inside the definition of (T*Q)(s,a), the probability measure P(s’|s,a) would be P^{(t)}.  Thus, T* is not one optimality operator shared amongst all tasks,  but one for each task.  The notation should reflect this.  Stating the descriptions of k, n explicitly in section 2 would be helpful.  I had to read to section 3 to be sure that k stands for a sequence of number 0 to big K and i stands for 1 to n with n being the number of samples.   Also, giving an intuitive explanation of gaussian complexity (1) and its use would also be helpful.

**Experience Assessment:**

I have read many papers in this area.

**Review Assessment: Checking Correctness Of Derivations And Theory:**

I assessed the sensibility of the derivations and theory.

**Review Assessment: Checking Correctness Of Experiments:**

I did not assess the experiments.

**Review Assessment: Thoroughness In Paper Reading:**

I read the paper at least twice and used my best judgement in assessing the paper.

---

> ### Author Response · Authors · 2019-11-08
> **Response to Official Blind Review #3**
>
> We thank the reviewer for the thoughtful comments. In the following, we address each point sequentially.
>
> Our architecture is inspired by the work in (Maurer, 2016), which was exploited to derive the multi-task approximation error bound. As reported in Section 2, page 4, of (Maurer, 2016): "The function $h: \mathcal{X} \to \mathbb{R}^K$ is called the representation, or feature-map, and it is used across different tasks, while $f$ is a function defined on $\mathbb{R}^K$, a predictor specialized to the task at hand.". As reported in (Maurer, 2016), e.g. in Section 5, a neural network with an arbitrary set of shared layers $h$ and different heads $f_t$, fits this mathematical framework. However, we consider RL tasks with different input spaces and the bound in (Maurer, 2016) cannot be applied directly. Thus, we derive an extension of the bound proposed in (Maurer, 2016) for different input spaces, and build the corresponding neural network architecture.
>
> Regarding the second concern: “Two, the analysis follows from an existing paper by Farahmand (2011), which has already established a bound on the difference $Q^* - Q^{\pi_K}$.  This paper considers the same difference, separately for each task, so that the same bound from Farahmand (2011) can be trivially used. The shared layer $h$ would affect the $Q$-values, but considering only the difference $Q_t^* - Q_t^{\pi_K}$ abstracts away how sharing representations is helpful. Simply averaging the norm of the difference between $Q_t^* - Q_t^{\pi_K}$ as if they are independent ignores the fact that the $Q_t$’s are all dependent due to the shared layer $h$.”. As pointed out in (Maurer, 2016) and in our Theorem 3, the upper bound of the approximation error $\varepsilon$ decreases for an increasing number of tasks, thanks to better feature extraction. The bound in Theorem 2 contains the task-averaged approximation error $\varepsilon_{\text{avg}_k}$, and therefore becomes smaller for an increasing number of tasks. The consequence is that a multi-task approach converges faster to the optimal $Q$-function w.r.t. the single-task approach. We hope we have solved this concern; otherwise, we will be glad if the reviewer can clarify his/her doubts.
>
> The approximation error $\varepsilon_k$ in (Farahmand, 2011) is definitely different from our task-averaged approximation error $\varepsilon_{\text{avg}_k}$, but the proof of Theorem 2 takes this into account. Indeed, Theorem 2 is the multi-task version of the AVI bound of (Farahamand, 2011), and it is, to the best of our knowledge, the first multi-task AVI bound. Note that it reduces to the (Farahamand, 2011) bound with a number of task $T = 1$. We can improve the description of the proof to make it clearer if the reviewer suggests it.
>
> The theoretical bounds are supported by the experimental results in Figure 1(b) and Figure 1(c).
> We consider the car-on-hill MDP, which is a not-trivial RL problem in which it is possible to compute the optimal $Q$-function. We show the norm-$1$ of the difference between the optimal $Q$-function and the $Q$-function at each policy update, exactly the value that we bound in Theorem 2. We show that considering multiple tasks is beneficial w.r.t. using a single task, since the convergence to the optimal $Q$-function is faster. The subsequent experiments are carried out on challenging RL problems in which the computation of the optimal $Q$-function is unfeasible, and therefore we focus only on performance. The performance of the multi-task approach is always better or equal to that of the single-task approach; however, note that even when these two approaches perform the same way, the multi-task is still more convenient because it uses fewer parameters than the overall single-task networks.
>
> Thanks for pointing out the notation issue about $T^*$. We uploaded a revised paper with the suggested fix. We also clarified the meaning of $n$ and $K$, and added an intuitive explanation of the meaning of the Gaussian complexity.

---

### Decision · Program_Chairs · 2019-12-19

**Decision:**

Accept (Poster)

**Comment:**

This paper considers the benefits of deep multi-task RL with shared representations, by deriving bounds for multi-task approximate value and policy iteration bounds. This shows both theoretically and empirically that shared representations across multiple tasks can outperform single task performance.

There were a number of minor concerns from the reviewers regarding relation to prior work and details of the analysis, but these were clarified in the discussion. This paper adds important theoretical analysis to the literature, and so I recommend it is accepted.